# Access to the Health Care System of Undocumented Moroccan Migrant Slum Dwellers in Southern Spain: A Qualitative Study

Fernando Jesus Plaza del Pino [1,†] , Ghita Chraibi [2] , Brigida Molina-Gallego [3,*] , Maria Humanes-García [3],
Maria Angustias Sánchez-Ojeda [4] and María Idoia Ugarte-Gurrutxaga [3,†]

1    Department of Nursing, Physiotherapy and Medicine, Faculty of Health Sciences, University of Almeria,
     04120 Almeria, Spain; ferplaza@ual.es
2    Centre for Migration Studies and Intercultural Relations, University of Almeria, 04120 Almeria, Spain;
     ghitachraibi9@gmail.com
3    Department of Nursing, Physiotherapy and Occupational Therapy, University of Castilla-La Mancha,
     45071 Toledo, Spain; maria.humanes@alu.uclm.es (M.H.-G.); maria.ugarte@uclm.es (M.I.U.-G.)
4    Department of Nursing, Faculty of Health Sciences of Melilla, University of Granada, 52017 Melilla, Spain;
     maso@ugr.es
*    Correspondence: brigida.molina@uclm.es; Tel.: +34-619085120
†    These authors contributed equally to this work.

**Abstract:** Spain has grown economically due to the recent boosts in the industrial sector, the agricultural sector, construction and services. Those who carry out agricultural tasks are mainly undocumented immigrants living in marginal neighborhoods. Objectives: to know the perception of undocumented Moroccan migrants living in marginal neighborhoods regarding access to the Public Health System. Methods: A qualitative method with a phenomenological approach was used to get closer to the experience of the participants in the study. In this work, 24 semi-structured interviews were conducted following a guide with a set of open questions to facilitate an in-depth discussion of the topics of interest. The participants were given an informed consent form, which guaranteed anonymity and confidentiality regarding the information obtained. For this, codes were used to identify them. The data were stored, managed, classified, and organized with the ATLAS-ti 9 software. Results: access and use of health services by the immigrants in the study was difficult due to communication problems related to language and culture, the discriminatory attitude and lack of cultural competence of health personnel, and the location of the neighborhoods (which were marginal far from urban centers), as well as the lack of transportation to health centers. Conclusions: it is an ethical imperative of the Health System to offer greater attention to the population in situations of extreme vulnerability, implement intercultural mediators and train health professionals in cultural competence.

**Keywords:** intercultural mediation; immigrants; cultural competence; health; qualitative research

## 1. Introduction

Human movement is a universal phenomenon that is inseparable from human nature [1]. We must keep in mind, as the World Health Organization indicates, that individuals who move are exposed to biological, psychological, and social risks that increase their situation of vulnerability: interruptions in health care, trouble accessing and using health systems, and inadequate treatments and increases in transmissible diseases [2]. In the last few decades, Spain has grown economically due to growth in the industrial sector, as well as the agricultural, construction, and service sectors.

According to the 2021 statistics, the number of foreigners in Spain ascended to 5,375,917, coming from more than 21 countries of origin [3]. In the study area, the foreign population is more than 20%, mostly Moroccans [4]. The concentration of immigration in southern Spain is tightly linked to agricultural activity in the province, as its continuous growth requires an abundant and cheap workforce [5].

Many of these immigrants are undocumented, which favors their access to precarious agricultural jobs, and many of them find themselves living in shantytowns that do not meet the minimum habitability conditions [6]. About 7000 undocumented immigrants live in this situation [7]. Workers in precarious employment situations often face greater demands or have less control over the work process and less experience. They suffer from social isolation and lack of support. These experiences have been identified as powerful social stressors, which in turn way be related to adverse health and well-being outcomes [8]. However, many studies, such as the one by García et al. on the settlements of temporary agricultural immigrant workers in Huelva, have provided evidence that the self-perception of health among these individuals is good and they do not often use health services, despite the conditions in which they live [6].

Generally, precarious jobs may limit worker's control over their professional and personal lives, resulting in experiences of job insecurity, feelings of betrayal and injustice, feelings of helplessness and lack of control, lack of future opportunities and lack of professional identity [9].

Precarious employment has been identified as a relevant social determinant of health and health inequalities. Certainly, this situation has a direct negative impact on their health and a strong impact on self-perceived health, including mental health [9–11]. The unhealthy living conditions in the shantytowns, and their repercussions on health physical, psychological, and social aspects, have been widely documented [12–14].

Universal health coverage is recognized in Spain, but there is a large body of academic literature showing that undocumented migrants face multiple barriers to access. [15,16]. Starting with the premise that health care must be addressed from a human rights approach [17]. We consider it very important to know the perception that these people have of the health care they receive.

The objective of this study was to understand the perception that Moroccan undocumented immigrants (UM) who live in shantytowns have regarding the Health System.

## 2. Materials and Methods

### 2.1. Study Design

A qualitative method with a phenomenological approach was used, as it is the most adequate method to achieve a profound and rich comprehension of the experiences from the perspectives of those who lived them [18]. In our case, the study was centered on the experience of UMs in specific living conditions (living in shantytowns in the south of Spain) in regard to the Health System. The recommendations of the COREQ guide for qualitative research reporting [19] were followed.

### 2.2. Participants and Research Design

In accordance with the objective of our study, we selected UM participants. The inclusion criteria were that the UMs resided in shantytowns in our area of study, rural areas of southern Spain, and who been in contact with the Health System. To obtain significant and generalizable results for the entire Moroccan population in this situation, purposeful sampling was used taking into account the following variables: sex, age, time spent in Spain and degree of knowledge of Spanish. The group of participants was composed of 11 men and 13 women (Table 1).

All of our informants had a temporary healthcare card that gave them access to the Public Health System.

The participants were recruited through collaboration with various members of NGOs that work in shantytowns, continuing with the snowball sampling technique to reach more participants. The participation was voluntary. Interviews were given until the research team considered that data saturation had been obtained [20], after which data collection ended.

**Table 1.** Characteristics of the participants.

| Code | Sex | Age | Time Living in Spain (Years) | Knowledge of Spanish (Yes/Little/None) |
|------|-----|-----|------------------------------|-----------------------------------------|
| M1 | Male | 29 | 1 year | none |
| M2 | Male | 23 | 1 year | little |
| M3 | Male | 22 | 3 years | none |
| M4 | Male | 27 | 5 years | little |
| M5 | Male | 33 | 4 years | none |
| M6 | Male | 32 | 3 years | none |
| M7 | Male | 30 | 4 years | none |
| M8 | Male | 28 | 5 years | none |
| M9 | Male | 30 | 4 years | none |
| M10 | Male | 32 | 8 years | yes |
| M11 | Male | 22 | 3 years | little |
| W1 | Female | 30 | 3 years | none |
| W2 | Female | 32 | 4 years | little |
| W3 | Female | 21 | 1 year | none |
| W4 | Female | 30 | 6 years | little |
| W5 | Female | 35 | 4 years | none |
| W6 | Female | 38 | 3 years | none |
| W7 | Female | 26 | 3 years | none |
| W8 | Female | 26 | 7 years | yes |
| W9 | Female | 25 | 5 years | yes |
| W10 | Female | 24 | 8 years | yes |
| W11 | Female | 32 | 4 years | little |
| W12 | Female | 37 | 2 years | none |
| W13 | Female | 31 | 3 years and 6 months | little |

### 2.3. Collection of Data

Data collection was conducted between January and April 2022, through semi-structured interviews following a guide with a set of open-ended questions to facilitate an in-depth discussion of the subjects of interest. All the interviews revolved around three phases: (1) presentation of the study objectives, (2) interview, in which the participants were asked about the living conditions in the shantytowns and their experience with the Health System, and (3) final phase, which included an open-ended question about proposals for improvements of health care for migrants in the Health System.

The interviews were conducted around the shantytown, in many cases in the house of the UM and, in others, in open areas, according to their preferences. The mean duration of the interview was 45-minutes and were conducted in Arabic by a Moroccan researcher. The interviews were recorded with the consent of the participants.

### 2.4. Analysis of Data

The transcription of the interviews was performed in Arabic by the Moroccan researcher, after which they were literally translated into Spanish. The data were stored, managed, classified, and organized with the ATLAS-ti 9 software. First, an iterative reading of the transcriptions was performed, followed by the identification of emergent themes and their classification [21], Each sentence from the transcription was analyzed line byline, and the researchers' identified units of meaning, sub-themes, and themes, which allowed us to understand the experience of the UM. The themes were initially aligned with aspects proposed in the interview, including sub-themes that emerged inductively.

### 2.5. Ethical Considerations

The study followed the ethical principles for human medical research adopted by the World Medical Association (WMA) in the latest declaration of Helsinki (64th General assembly, Fortaleza, Brazil, October 2013). The Research Ethics Committee from the Department of Nursing, Physical Therapy, and Medicine, University of Almeria, approved the research pro-

tocol (Registration number: EFM 162/2021) for the development of the study. An explanation of the objective of the study was given to all the participants, as well as the informed consent form, which shows the guarantee of anonymity, and the confidentiality of the information obtained. All the participants signed the form. To ensure the anonymity and confidentiality of the participants, codes were utilized to identify each of them.

## 3. Results

After identifying forty-two codes, the most important ones were selected to group the data into key categories in order to focus on the most relevant aspects of our study. Four categories were finally established.

Table 2 contains a summary of the categories, sub-categories, and codes that appeared from the content analysis in line with the different questions posed to the informants.

**Table 2.** Categories, sub-categories and codes.

| Categories | Sub-Categories | Codes |
|---|---|---|
| **Living in the shantytown** | **Health in danger** | Living in extreme poverty |
| | **Added risks for women** | Sexual harassment and abuses<br>Need for protection |
| | **Isolation, another barrier** | With a healthcare cared but forgotten<br>Far from urban areas |
| **Experiences in the health system** | **Use of the health services** | Proof of integration<br>Health problems |
| | **Quality of care** | Standard treatment<br>With COVID-19, worse care<br>Positive experiences |
| **The importance of communication in the Health System** | **Language as a barrier** | Communication problems<br>Clinical disinformation<br>Bureaucratic disinformation |
| | **Solutions for the communication problems** | Non-verbal language<br>Informal translators<br>Formal translators<br>Google Translate<br>Independence after learning the language |
| **Health professionals and health institutions** | **Discrimination and cultural awareness of health professionals** | Discrimination and lack of professionality<br>Positive experiences<br>Lack of cultural awareness of the health personnel<br>Cultural awareness of the health personnel |
| | **Intercultural mediation as a proposal for improvement** | Opinions about intercultural mediation |

### 3.1. Living in the Shantytown

The first part of the interviews delved into life in the shantytowns, focusing on the repercussions to their health and the access to the Health System, aspects of great interest for coming close to the daily reality of these individuals and to contextualize our results. The conditions for the UMs in shantytowns are extremely precarious, as shown by the lack of resources, the characteristics of the shacks, the surroundings, and the physical and social isolation they suffer. These circumstances negatively affect their health and have become other barriers against access to health services.

### 3.1.1. Health in Danger

Most of the informants were healthy, although they suffer continuous stress that affects their general well-being, related to the conditions of extreme poverty in which they live.

*I don't have any illnesses; I don't have anything. What I have is that I don't have work or nothing. I worry a lot, I don't sleep well, I keep everything inside, and my heart hurts; I have a family and I have to send them money. W7*

*Living under plastic is not living, when it's hot, you can't be inside, and when it's cold, it's too cold. There is no water, some days we have power, and in others, we don't. W3*

### 3.1.2. Added Risks for Women

Aside from suffering from these living conditions, they are continuously exposed to sexual harassments and abuses by men inside and outside the settlements.

*Now if you ask for work, the boss tells you to go with him, he asks that you become his prostitute, and my sister, this is not for me. W8*

Some decide to have a partner in order to have economic support and protection against the possible threats.

*In this country, if you don't have a boyfriend to take you to the doctor, who's going to take care of you or give you money? Because work for being able to live every day does not exist, us women when we are alone, we suffer a lot, when men see you alone, they take advantage of you, and surpass the limits. W9*

### 3.1.3. Isolation, Another Barrier

The UMs who participated in the study have healthcare cards thanks to the work of the NGOs that work with it. However, despite being users of Public Health and finding themselves in a situation of extreme vulnerability, there has been a lack of follow-ups by the Health System or Social Services in the settlements, even during the pandemic.

*No one comes here, we only get help from volunteers from associations; they obtained a card for the doctor. M4*

*If you become sick, and you can't move, you have to be clear that although you call the health center, no doctors are going to come, no one has ever come. M10*

*We got the COVID vaccine from the Red Cross, they also gave us masks, although living here without water... M9*

The shantytowns are located outside urban centers, close to the greenhouses but far from public transport. This makes it difficult to access health services.

*They take us by car for 25 Euros, because the bus doesn't get here and it's very far, we don't know where we can get the money, if we don't have it, we stay in our shacks until we get better. M11*

### 3.2. Experiences in the Health System

This category encompasses the comments expressed by the UMs about their experiences in the Health System, including their use of the public health resources and how they perceive the care received by health professionals, with frequent references to nurses.

### 3.2.1. Use of the Health Services

The UMs indicated that they go to health services sporadically to solve occasional health problems they may have. The frequency is very low.

*Since I got to Spain seven years ago, I've gone to the doctor's four times...I haven't needed much, really. W8*

3.2.2. Quality of Care

The participants perceived bad quality care in the doctor consultations, stating that there was a lack of dedication to the patient. They were all offered practically the same treatment without diagnostic tests or physical examination.

*They give everyone a blue pill, for anything, that blue pill, everyone that goes, the same for all, no analysis, no X-rays, nothing. M9*

This situation is attributed to the language barrier, which impedes them from expressing their symptoms or participating in the medical decisions.

*Here, you can only tell the doctors that your head hurts, your stomach hurts, you can't tell them anything else…if you want to tell them about other problems, you can't. M7*

*When the nurse comes, I have to agree with whatever she wants to do, because I don't understand what she says. W7*

With the arrival of COVID-19, access to health services was limited, entrance to the patient's companions was denied, and most of the medical consultations were made through the phone. For those who do not speak Spanish, these changes made communication with health personnel impossible, with some mentioning going to emergencies so that they could be attended in person.

*Now only with the cellphone, now people cannot go, the translation is not there, and you can't understand anything through the cellphone. M6*

*I had asthma, chest pain, when I have something I go directly to emergencies, I don't go to the health center anymore since coronavirus started because they gave me a phone appointment….and of course, and I don't understand anything. W2*

We also found UMs who had positive experiences in their contact with the Health System.

*They treat us well, just as the Spanish people, the nurse sit with me, she give me time, and treat me with a lot of patience. W4*

*I go due to my diabetes, my nurse explains everything well, motivate me to improve…I'm very happy. M11*

*3.3. The Importance of Communication in the Health System*

The following category encompasses the difficulties that UMs found related with communication in the different health services and the manners in which this communication problem could be solved.

3.3.1. Language as a Barrier

The participants coincided in that the language barrier was an obstacle against their access to quality health care. They are aware of the importance of language and the problem that this barrier could become in their relations with health personnel.

*There are no confusions…I say "very nervous" and "head hurts", and it is understood (laughs), this is the problem, I say "very nervous" and that's it, I don't give any more details, how can I say it if I don't know how it's said? W5*

The patient's right to information during the clinical process is not respected, many informants stated that they do not receive any explanations about their disease or the diagnostic tests they are subjected to.

*They give you the prescription, but they don't explain anything…and if you don't understand, it's your problem. M4*

*They don't show you the radiographs, they don't tell you anything, only the pill, and that's it. The pill doesn't do anything. M5*

The administration procedures or the appointment management system of the health centers are very complex, and our informants confess to not knowing how to ask for an appointment. The fact of not knowing the language complicates this situation even further.

*I look for appointments in the application, but they give you one in a week or two...but, if it's an emergency, what should I do? W8*

*I've gone many times...I wait there and then I come back...I don't know what I have to do, and no one explains anything. W3*

3.3.2. Solutions for the Communication Problems

The participants emphasize the need for a translation service within the hospitals and health centers.

*When you ask for a translation, they don't bring translators, there aren't any. M9*

*On many occasions, we need to go to the doctor, but the person who translates for us is working or has other things to do, and in the end we don't go, or if we go, we don't understand anything. M1*

The UM mentioned that only one hospital had a hired translator; in other centers, translation is offered through the phone, although they also mentioned that they only have access to formal translation in some cases.

*Sometimes when it's very important, they call the translator. M5*

*If you insist and say, "I don't understand anything, I want a translator", they call on the phone and translate everything for you. W2*

Given the absence of formal translators, the participants resort to the use of non-verbal language with a few words in Spanish they learned, without being sure that the message is correctly understood.

*When I arrived, I said "little pee pee, very little, and much hurt", pointing at myself...he understood!...and that's it. W4*

*I talk with my hands, and if needed, with my feet, so that they understand. M6*

They also resort to family members and friends for translation. When people close to them are not available, they are forced to ask for help from unknown individuals in the waiting room, renouncing their right to intimacy and confidentiality.

*I take someone for translation, and I finish sooner...although maybe there are personal things that I don't want to tell, but I do it...this is better than to have something happen to me. W6*

*When we go, sometimes we find someone in the room, and we ask them to go in with us to the consultation. W2*

Some participants are able to use new technologies use tools such as Google Translate to communicate with the health personnel or to translate medical reports.

*Those who know a little, translate it to Arabic with the cellphone. W6*

*Something happened to me once, and I did not want anyone to know, I used Google translate and it went well. M3*

For the UMs who speak Spanish, it is completely different, as they can be more independent when going to health services. They are able to establish adequate communication with the health personnel and perceive better health services.

*Since I arrived here in Spain, I've had many misunderstandings, before I used to go with a friend who speaks Spanish, now I understand by myself, I go to the doctor by myself...I give my name, they give me a number, and I go in normally. W9*

*3.4. Health Professionals and Health Institutions*

In this category, we examine the perception of the participants regarding the cultural sensitivity of health professionals towards people of Moroccan origin and the discrimination situations the participants have experienced. At the same time, it is established that intercultural mediation is a necessity for improving the health care of immigrants.

3.4.1. Discrimination and Cultural Awareness of Health Professionals

The UMs indicate having to face situations of discrimination by health personnel. Those who do not speak Spanish well tend not to report these behaviors. However, those who do speak Spanish are able to defend themselves to obtain better care.

*There is a lot of discrimination. I've been told many times that nothing hurts and that I'm lying, they tell it to your face, as since you can't say anything, you leave. You don't speak the language, and you are also afraid that they will call the police, we don't want trouble. You just take it and leave. M5*

*The orderly who pushed my wheelchair was saying "these people come and perform theatrics", she goes in with the nurses and I hear her say "here they say they don't speak (Spanish), and then in the markets, they know how to, she kept on saying atrocities, I went to write a complaint, I think she treats all Moroccans the same, and of course, no one says anything. W10*

The UMs perceive a lack of cultural awareness and lack of knowledge of the personnel due to their religious beliefs and practices.

*He told me during Ramadan "if you feel bad, drink water, why do you come here? She doesn't know anything about Ramadan. . .he says drink water and that's it. M8*

*You feel that they don't understand you. . .on many occasions, I've been attended by male doctors, they have examined my private parts, I felt really ashamed. . .they should have told me what they were going to do, and who was going to do it. W4*

Some participants mention finding health professionals who were respectful and aware of cultural differences.

*I had an accident, and they gave me pills, I explained to them that it was Ramadan, and they told me to take them once fasting ended. . .they understood that during the day, I couldn't take pills. M5*

3.4.2. Intercultural Mediation

The UMs suggested that it would be very helpful to have translators in the Health System enacting functions that could correspond with those of an intercultural mediator, such as translation, cultural bridging, etc. Although the informants already knew of the existence of this individual, who had functions similar to those they discussed.

*Some things the doctor says you can't understand even though you know the language. . .if there was someone there who knew about the subjects, it would be better. M8*

*It is very important for Moroccans to be there to understand everything, without any confusion, and that they know about our customs. I hope they hire even Moroccan doctors. M4*

**4. Discussion**

The results of our study highlight the deplorable state of the mental health of those living in the settlements, in a situation of poverty and under the pressure of an additional burden related to the survival of their families in their countries of origin. In a study conducted in Canada with a migrant population, emphasis was placed on the need to maintain a family as one of the reasons for suffering the work risks in silence, as well as maintaining the sole source of income [22]. The pressure due to the family burdens, and its consequence on mental health, was reflected in studies similar to the present one [7,23,24].

This situation is even worse for women, who also mentioned that they are constantly subjected to sexual harassment and abuse by men both inside and outside the settlements. We can attest that, all over the world, female rural workers live in a situation of symbolic and habitual violence because they are women, which manifests itself in harassment and aggression based on gender inequalities [25]. The power relationship between men and women is a constant in all spheres, and one of its manifestations is the violence perpetrated against them simply because they are women. The intersection of gender with other structural social determinants such as territory (rurality), social class (poverty) and working conditions, which come together for women living in settlements, places them in a situation of greater vulnerability than men.

The incorporation of the gender perspective and the intersectional approach in the analysis of the reality of the women in our study allows us to understand the processes of socio-cultural construction of feminine and masculine subjectivities in a common way, based on power inequalities and social relations of domination and subordination [26], as well as the different models of oppression [27].

On the other hand, the location of the settlements where the people we interviewed live, in areas close to farms and without transport services, is one of the aspects discussed in the study as a difficulty in accessing some health-related services. This situation was also mentioned in a report by the Spanish government in 2022 [28]. Another aspect of health care that emerges from the findings is that, although all UMs in the study have a health card and therefore have a recognized right to public health care and related services, they feel abandoned by the Health System. They mention that "home" care is not possible although they may need it, and those who provide this care are third party organizations.

With respect to the quality of care perceived, most of the study participants complained about the standardized and non-personalized care, which they blamed on the language difficulties [29]. This could be in relation to the technical language utilized in healthcare, which comes from the biomedical approach that still prevails in the system [30]. This bias has especially negative consequences when discussing the care provided to individuals who have very different cultures, languages and beliefs.

This situation worsened during the COVID-19 pandemic. In the overcrowded conditions of the shantytowns, it was very difficult to respect the social distance and other basic prevention practices, such as self-isolation, in cases of infection.

The language barrier was identified as the main obstacle in the access to quality healthcare and the cause of problems participants had in their relationship with the health professionals. They complained about the infringement of the rights that they, as patients, have for receiving information about their disease and the diagnostic and therapeutic proposals. The proposals discussed to solve the problem included the implementation of a translation service. When faced with the absence of formal translators, they rely on the use of non-verbal language with the use of a few words they already know in Spanish, which has proven insufficient for establishing effective communication. The need for adequate communication between patients and professionals has also been broadly described in the literature as one of the main challenges for the effective access of the migrants to health care [31–33].

To overcome this barrier, the UMs resort to informal translators. This type of practice, aside from infringing on their rights, as it violates confidentiality [34] in the management of health information and independence in the making of decisions, can lead to diagnostic and treatment errors [35–38]. The practice of good communication skills in the area of health is fundamental to the development of a significant and trustworthy relationship between health professionals and patients, and therefore it is beneficial for both [39].

Our participants experienced discriminatory treatment by health professionals, which they attributed to racism, lack of knowledge about their customs and lack of respect for their cultural practices. This perceived discrimination is common in studies similar to ours [40]. This perception could be mediated by misunderstandings due to the difficulty in communication, and there are also more profound conditioning factors associated with

the values and beliefs of both parties [41]. The feeling of uneasiness of our patients goes against the principle of humanization in health care, which includes, among others, key concepts such as patient satisfaction, patient-centered care, and the participation of the patient. However, the efforts made to identify the "experience of the patient" have normally only explored the opinions of patients who belong to dominant groups [42].

The existence of a person that knows both the language and the customs of the culture is one of the demands we have found. This profile corresponds to intercultural mediation in the area of health. In this sense, many studies have also described the same results [33,40,43,44].

In this sense, we share the idea that training health professionals in culturally competent care is necessary in the approach to health care for this population, as advocated by other studies that highlight the difficulties resulting from the cultural distance between patients and professionals [45].

*Limitations of the Study*

Some of the limitations that we want to describe are related to the characteristics of the sample. The scarce linguistic competence in Spanish of a great part of the sample of Moroccans who were initially selected led to the use of a single interviewer, a researcher from our team whose mother tongue is Moroccan Arabic, which led to additional limitations.

As for the methodology, it is necessary to have in mind that the study was centered on shantytowns in a specific area of Spain. Although it is very similar to other areas, and the sample was chosen to be representative of this reality, we cannot assume that the results from this work can be directly applied or extrapolated to every person who lives in other settlements.

Nevertheless, we must remember that the qualitative study is not based on statistical representation of the sample, and the intention is not to achieve the representation of a population based on its conclusions. It must be remembered that each methodology comes with certain cautions in the use of the findings.

Another limitation was related to the procedure, particularly aspects associated with the manner in which the information was collected and the interaction between individuals when giving an interview. The type of relationship that is established can condition the collection of data, either because the characteristics of the interviewer have an influence on the participant or because communication problems are produced.

**5. Conclusions**

It is an ethical imperative of the Health System to offer greater care to the population in a situation of extreme vulnerability, as is the case of UMs living in shantytowns, especially women.

Most of the participants had a bad perception of the health care they received, associating it with abandonment, lack of empathy and respect. They believed that they are not listened to, or that their right to health care is not respected, despite it being legally recognized with the possession of a healthcare card.

In order to guarantee quality care for UMs, it is necessary to implement intercultural translators/mediators in the Health System, as well as training health professionals in cultural competence.

**Author Contributions:** Conceptualization, M.I.U.-G. and F.J.P.d.P.; methodology, F.J.P.d.P.; M.I.U.-G. and F.J.P.d.P. validation; formal analysis, G.C., M.I.U.-G., M.H.-G. and M.A.S.-O.; research, G.C., M.I.U.-G., B.M.-G. and F.J.P.d.P.; writing: preparation of the original draft, G.C., M.I.U.-G. and F.J.P.d.P. writing: review and editing, G.C., M.I.U.-G., M.H.-G. and F.J.P.d.P.; supervision, F.J.P.d.P. All authors have read and agreed to the published version of the manuscript.

**Funding:** This research did not receive external funding.

**Institutional Review Board Statement:** The study followed the ethical principles for human medical research adopted by the World Medical Association (WMA) in the latest declaration of Helsinki

(64th General assembly, Fortaleza, Brazil, October 2013). The Research Ethics Committee from the Department of Nursing, Physical Therapy, and Medicine, University of Almeria, approved the research protocol (Registration number: EFM 162/2021) for the development of the study.

**Informed Consent Statement:** Informed consent was obtained from all subjects involved in the study.

**Data Availability Statement:** The data presented in this study are available on request from the corresponding author.

**Public Involvement Statement:** No public involvement in any aspect of this research.

**Guidelines and Standards Statement:** This manuscript was written following the recommendations of the COREQ guide for qualitative research reporting [19].

**Acknowledgments:** The authors thank all migrants who participated in this study.

**Conflicts of Interest:** The authors declare that they have no conflicts of interest.

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
