# Peer review of "Access to the Health Care System of Undocumented Moroccan Migrant Slum Dwellers in Southern Spain: A Qualitative Study"

_nursrep, doi:10.3390/nursrep14010038_

Round 1

Reviewer 1 Report

Comments and Suggestions for Authors

Comments on the Quality of English Language

Quality of English is good.

Author Response

We attach the document that includes the suggestions from the previous review.

Reviewer 2 Report

Comments and Suggestions for Authors

The article addresses an important issue in the right way in a well-conducted qualitative study. There are important points to consider, which I'll list below. I have attached the text document, where I have made some notes and direct reminders to the authors.

1. the title of the article should be reconsidered. According to what is presented in the objectives and the data collected, the title should be  something like - "The perceptions of Moroccan undocumented immigrants slum dwellers about the health care system: a qualitative study".

2. An important question in the paper is whether they really want to assume that this study is phenomenological. What is presented here is a qualitative study with semi-structured interviews. Nowhere do they justify the study as phenomenological, which I don't think it is.

3. In the introduction, the number of participants is wrong. According to the table above, there are a total of 25. Table 1 must be improved.

3. In view of the interviewees' difficulties with the language,  it should be explained how the interviews were conducted.  If a person to translate has present, if they used machine translation....

4. Please check all the notes in the document. Some specific aspects of the English should be improved.

Comments on the Quality of English Language

Although I don't have all the knowledge to assess this point, the language seems correct to me. There are occasional expressions that need to be revised and improved. Also, especially in the discussion and conclusions, more academic English should be used, in the sense of not being so affirmative, since we are in a partial, qualitative study and should therefore use more scientific language.

Author Response

(The authors gave the same response as above.)
